# Impact of Variation between Assays and Reference Intervals in the Diagnosis of Endocrine Disorders

**DOI:** 10.3390/diagnostics13223453

**Published:** 2023-11-16

**Authors:** Nathan Lorde, Ahmed Elgharably, Tejas Kalaria

**Affiliations:** Black Country Pathology Services, The Royal Wolverhampton NHS Trust, Wolverhampton WV10 0QP, UK; ahmed.elgharably2@nhs.net (A.E.); tejaskumar.kalaria@nhs.net (T.K.)

**Keywords:** endocrine disorders, method comparison, reference interval, discordance, harmonisation, standardisation

## Abstract

Method-related variations in the measurement of hormones and the reference intervals used in the clinical laboratory can have a significant, but often under-appreciated, impact on the diagnosis and management of endocrine disorders. This variation in laboratory practice has the potential to lead to an errant approach to patient care and thus could cause harm. It may also be the source of confusion or result in excessive or inadequate investigation. It is important that laboratory professionals and clinicians know about these impacts, their sources, and how to detect and mitigate them when they do arise. In this review article, we describe the historical and scientific context from which inconsistency in the clinical laboratory arises. Examples from the published literature of the impact of the method, reference interval, and clinical decision threshold-related discordances on the assessment and monitoring of various endocrine disorders are discussed to illustrate the sources, causes, and effects of this variability. Its potential impact on the evaluation of growth hormone deficiency and excess, thyroid and parathyroid disorders, hyperandrogenism, hypogonadism, glucocorticoid excess and deficiency, and diabetes mellitus is elaborated. Strategies for assessment and mitigation of the discordance are discussed. The clinical laboratory has a responsibility to recognise and address these issues, and although a lot has been accomplished in this area already, there remains more to be done.

## 1. Introduction

Most laboratory assays were initially developed inhouse by different laboratories and, in the case of endocrine assays, mainly since the mid-twentieth century. The blood and fluid evaluations were employed for local populations, and the generated patient results were compared with inconsistently defined values called “normal ranges”. It was realised with time that multiple different “normal ranges” were required for different populations and laboratories because of differences in demographics and methods [1,2]. The concept of the “normal value”’ was challenged by Gräsbeck and colleagues [3] because the distribution of most biological analytes is non-Gaussian. They coined the term “reference interval” to describe fluctuations in analyte concentrations in well-characterised groups of individuals [3]. It was, and often is, assumed that a “normal” result means that the patient has no pathological derangement. The term “reference interval”, however, better denotes that the intent is to provide a reference against which individual results are compared and therefore it is more meaningful in conveying the continuous, rather than binary, nature of the clinical risk [2]. Additionally, clinical decision limits, general or situation specific, are increasingly employed to identify results requiring an action, for example, further investigations, an intervention, or a change in treatment [2].

Variation in performance characteristics across laboratories, as well as in reference ranges used for analytes, can be problematic. The reasons are complex and multifactorial. Often this variation is overlooked or ignored, probably because it is difficult to identify and even more so to correct. No set of disorders is affected by this at the scale of the pathologies of the endocrine system, whose diagnosis and management rely heavily on results of biochemistry tests. The following examples are but some of the analytes whose variability in assay and reference intervals poses significant challenges for the diagnosis and management of disorders of the endocrine systems. The effects of cross-reactivity and interference in hormone immunoassay are discussed at length in a relatively recent review by Ghazal et al. [4] and therefore we have not covered those aspects in this review [4].

## 2. Aim and Method

We aim to highlight the differences between hormone assays and their reference intervals, and their potential impact on the management of endocrine disorders. The supporting literature for this narrative review was searched using PubMed.

## 3. Condition-Specific Examples of Assay and Reference Interval Discordance in Endocrine Disorders

### 3.1. Growth Hormone (GH) Deficiency and Excess

Insulin-like growth factor 1 (IGF-1) measurement is critical for the evaluation of somatotropic axis disorders. IGF-1 measurement is preferred to the measurement of GH because a randomly taken sample for GH has a large intra-individual variation, both in health and disease states. Additionally, many other factors, besides pathological pituitary production of GH, can lead to alterations in blood concentrations of GH, for example, malnutrition, renal disease, and liver disease [5]. IGF-1 levels, on the other hand, are controlled by the overall average of the daily GH secretion and provide a better idea of aberrations in GH secretion by the pituitary. However, different IGF-1 assays give differing results, generally believed to be because of differences in calibration as well as varying efficacy of IGF binding protein removal prior to measurement [6,7]. IGF-1 changes significantly with age, necessitating multiple age partitions of reference intervals. Additionally, IGF-1 is affected by multiple other factors that make defining a reference population challenging [7]. Studies have shown discordant IGF-1 and GH interpretations using manufacturer-provided reference intervals in GH deficiency and excess [8,9,10]. However, reference values derived from large numbers of age-adjusted controls provided correct interpretation of IGF-1 results in treated patients with symptomatic GH excess in one study [8]. IGF-1 reference intervals derived for six immunoassays in another study demonstrated generally poor concordance with their corresponding manufacturer-supplied reference intervals, moderate to good agreement among the results of the assays, but significant differences in reference intervals for each assay when derived from the same large reference population [6]. This study highlighted the need to derive IGF-1 reference intervals from a large reference population for each assay and how vital it is to use the same assay in serial monitoring of patients. In practice, an IGF-1 result marginally outside the assay-specific age- and sex-related reference interval may be because of a step change in the reference interval from one age bracket to the next. When interpretating IGF-1 results, the change with age should be considered to be continuous rather than stepwise, and it may help to consider the result in the context of the next higher or lower age bracket reference interval in addition to the patient’s age bracket.

GH measurements are used in dynamic function tests to confirm and monitor somatotropic axis disorders. Failure to suppress GH levels after an oral glucose tolerance test can be confirmatory of excess GH secretion, and failure of GH levels to rise sufficiently after an injection of glucagon strongly suggests GH deficiency [5,11]. However, discrepancies between results of GH dynamic function tests and IGF-1 levels are known to occur, and this can pose a particular challenge in monitoring patients with GH excess who are receiving treatment or have had definitive pituitary surgery previously [12]. This may be due to a number of factors: the disease process itself; other patient factors that impact GH levels, such as malnutrition, diabetes mellitus, thyroid disorders, kidney or liver failure, or adolescence; or, importantly, inappropriate cut-offs for GH levels in the dynamic function tests and/or IGF-1 assays used, or the performance characteristics of the assays [6,7,12,13,14]. Specifically, for IGF-1, the variable ability of different immunoassay kits to separate the hormone from its binding proteins is a known cause of differences in measurement capabilities of different assays [15]. This situation, understandably, can cause significant difficulty for the treating clinician.

### 3.2. Thyroid Disorders

Thyroid disorders, hypothyroidism in particular, are common. Measurement of thyroid stimulating hormone (TSH), the hormone secreted by the anterior pituitary that stimulates the thyroid to make and release thyroid hormones, is the usual initial test for disorders of the thyroid axis, either on its own or with measurement of free thyroxine (fT4). TSH is under negative feedback control by the thyroid hormones thyroxine (T4) and triiodothyronine (T3). TSH is very sensitive to the output of thyroid hormones and its concentration will change dramatically in response to subtle changes in the concentrations of thyroid hormones, a fact that makes it very suitable as an initial test [16]. Due to the negative feedback control system, TSH will rise in primary thyroid failure and will fall during primary hyperthyroidism. Care has to be taken in certain situations when interpreting TSH results, however. Pituitary disease, namely pathological over- or under-secretion of TSH driving hyper- or hypothyroidism, respectively, will obviously cause a TSH result that is the opposite of what is usually seen in primary thyroid diseases. If a clinician is suspecting these secondary (pituitary driven) forms of hyper- and hypothyroidism, then the free thyroid hormones will need to be assayed along with TSH. The first trimester of pregnancy is also special in requiring measurement of free thyroid hormone along with TSH because human chorionic gonadotropin (hCG) secreted by the placenta can stimulate the TSH receptor on the thyroid gland and cause release of thyroid hormones. The negative feedback that results from this leads to a suppression of TSH release from the pituitary—a picture that would look like primary hyperthyroidism if only the TSH is measured.

When the thyroid hormones (T3 and T4) are measured directly, it is the unbound version that is usually measured, hence the terms free T3 (fT3) and free T4 (fT4). This is because these hormones are heavily bound in serum to thyroid binding globulin, trans-thyretin, and albumin, by up to 99.97% [16], but it is thought only a small amount of free hormone is able to enter cells and stimulate the thyroid receptors. The concentration of the total hormone, either T3 or T4, will be susceptible to changes in serum proteins, which themselves can be affected by other factors such as nutritional status, inflammatory states, liver disease, sex hormone levels, and some drugs, but which will not result in changes to the concentrations of the free forms of the hormones available to enter cells [16].

Subclinical hypothyroidism affects up to 10% of the population and the management guidelines recommend regular monitoring and levothyroxine replacement if the TSH rises to ≥10 mIU/L or at lower TSH values if the patient is symptomatic [16,17,18,19]. However, despite the work by the International Federation of Clinical Chemistry and Laboratory Medicine (IFCC) working group for the standardisation of thyroid function tests (C-STFT) [20,21], TSH and fT4 immunoassays in routine use are not fully harmonised yet [20,21]. Studies have identified proportionate bias in Abbott’s and Roche’s TSH and fT4 assays [22,23]. In a recent study assessing clinical management discordance of subclinical hypothyroidism using Abbott’s and Roche’s TSH and fT4 assays, median TSH and fT4 results on the Roche platform were 40% and 16% higher than Abbott’s results, respectively. However, despite higher results, the upper reference limit provided by Roche for TSH is lower than that for Abbott and the lower reference limit of Roche’s fT4 is higher than that of Abbott. The combination of assay bias and differences in the manufacturer-provided reference intervals led to a substantial discordance in the diagnosis and management of subclinical hypothyroidism. Of 40 consecutively identified subclinical hypothyroidism patients using Abbott’s assays and 53 using Roche’s assays, only 41 (44%) had concordant diagnoses of subclinical hypothyroidism requiring observation across both platforms [24]. A similar study, which assessed levothyroxine dosage decisions in 100 consecutive patients with primary hypothyroidism, identified a potential 14% difference in decision making when comparing Abbott’s and Roche’s TSH assays [25].

These studies highlight method-related differences that are not accounted for by the widely used manufacturer-provided assay-specific reference intervals. These differences are not considered in clinical management guidelines and will lead to differing management decisions depending on the assays used. Of particular importance is the decision threshold of TSH ≥10 mIU/L in subclinical hypothyroidism for levothyroxine initiation [24].

### 3.3. Polycystic Ovary Syndrome

Polycystic ovary syndrome (PCOS) is a heterogenous condition affecting women of reproductive age, and one of its defining features is clinical or biochemical hyperandrogenism. The 2023 International Evidence-Based Guideline for the assessment and management of polycystic ovary syndrome [26] states, with respect to measuring androgens in females, that “reference ranges for different methods and laboratories vary widely” and that they are likely derived from groups that include women with PCOS [26]. The unintended consequence of this is that diagnosis rates of PCOS can vary depending on the reference values of the laboratory used to aid diagnosis. The guideline goes on to recommend use of liquid chromatography-tandem mass spectrometry (LC-MS/MS) for measurement of testosterone in females, stating that immunoassays do not have the desired accuracy, sensitivity, or precision for diagnosing hyperandrogenism in women with PCOS [26]. Thus, not only does variation in assay performance and reference interval impact the diagnostic difficulty in this condition in some women, but the use of poorly suited methods may also compound this further. The guideline recommends that androstenedione and dehydroepiandrosterone sulphate (DHEAS) can be measured in those for whom PCOS is clinically suspected but whose testosterone is not elevated, again recommending LC-MS/MS as the method of choice [26].

### 3.4. Hypogonadism

Hypogonadism is the pathological state when the gonads, i.e., the ovaries in women and the testes in men, are not functioning effectively or, in severe cases, at all. Both ovaries and testes make hormones which have vital roles all over the body. Hypogonadism may be due to pathology in the gonads themselves or may be secondary to deficiency in the gonadotrophins, the hormones released from the anterior pituitary that stimulate the gonads to make the sex hormones and perform some of their other roles.

Testosterone is the principal hormone released from the testes in men and measurement of its concentration in blood is used as the main test for male hypogonadism. Testosterone levels are higher in males than in females. The measurement of testosterone in men is therefore not thought to be as technically challenging as in females, as we referred to in the above section on PCOS. However, an audit of 96 British laboratories showed a large variation in the measurement of male testosterone concentrations [27]. Lower reference limits quoted by the laboratories ranged from 5.0 to 11.0 nmol/L. A portion of 50% of these laboratories obtained their reference intervals from the manufacturer, while only 8.3% created their own reference intervals from their local population. An external quality assurance (EQA) distribution of testosterone results in the same study showed the analytical variation present in British laboratories, especially among immunoassay platforms, with some labs having >25% bias from the target. Most of the laboratories used immunoassays and large differences were seen even among users of the same immunoassay platform [27]. This discrepancy in results and reference ranges will mean that some men may experience disparity in diagnosis and/or treatment of hypogonadism, especially those with borderline-low testosterone levels.

Female hypogonadism may be even more difficult to diagnose and monitor than in men. Due to the cyclical nature of the menstrual cycle, concentrations of hormones made by the ovaries change significantly throughout the month in a healthy woman. Reference ranges for oestradiol and preogesterone have been made for the different stages of the cycle. However, accurate assignment of the correct range depends on knowing which stage the woman is in at the time of phlebotomy, and this can be difficult as some forms of ovarian failure cause abnormal cycle lengths and frequency. Additionally, there are the cases where one wishes to evaluate a woman who has not yet experienced menarche for ovarian function. Measurement of the pituitary gonadtrophins may be helpful in some cases.

The laboratory-related variation adds to this complexity. Oestradiol measurement using immunoassays often has a positive bias compared with LC-MS/MS, thought to be due to interference by other oestrogen metabolites [28]. The Endocrine Society [29] issued a position statement on the measurement of oestradiol and in it included the fact that at low concentrations, seen in children, post-menopausal women, women being treated for breast cancer and men, oestradiol is not easily and accurately determined by most modern methods with good precision [29]. A 2007 study cited by the Endocrine Society in their position statement showed that Belgian laboratories in an (EQA) scheme showed bias from target of −26 to +239% for oestradiol and −23 to +81% for progesterone [30]. Another study comparing oestradiol measurements on different platforms found that some methods could report values as much as 6 times the value reported on another platform for the same specimen [31]. One sample in that study had measured values that ranged from 9.4 pg/mL to 64.8 pg/mL across different analytical platforms, where the target result was 14.1 pg/mL [31]. The authors highlighted the significance of this difference by pointing out that the European Menopause and Andropause Society [32] advised an oestradiol concentration less than 14 pg/mL would be consistent with a diagnosis of premature ovarian failure if the FSH is above 40 IU/L in the right clinical context [32]. It is thus easy to see how laboratory analytical discrepancies can make it difficult to diagnose a woman with hypogonadism, especially if cut-off values not guided by the local laboratory’s analytical performance are used.

### 3.5. Parathyroid Disorders

Parathyroid hormone (PTH) currently is routinely measured using second-generation, and in some laboratories, third-generation immunoassays. These immunoassays are made using pairs of antibodies with the intention that the entire PTH molecule alone, rather than circulating fragments, causes a measurable signal. The aim of this was to reduce the positive bias seen in first-generation assays caused by fragments of PTH. However, when three different commercially available platforms were used to compare parathyroid, calcium, and albumin results in patients with possible normocalcaemic primary hyperparathyroidism (NCPHPT) in a recently published study, there was surprising incongruity. Fifty-five consecutive samples indicative of NCPHPT in one lab were included in the study. Of those, only 29% had the same interpretation in the second lab and 20% in the third [33]. This variation was largely due to variation in the PTH results and reference intervals used by the different labs, but there was also a discrepancy in calcium that contributed. These three laboratories were close geographically, so a patient’s diagnosis could potentially change from “normal” (calcium and PTH within their reference intervals), to NCPHPT or frank primary (hypercalcaemic) hyperparathyroidism, depending on whether their blood was tested at one of the three sites [33]. It is thought that newer-generation PTH assays still suffer variable interference by fragments and some have turned to the development of LC-MS/MS methods that may, however, not be suitable for routine use for a high-throughput test such as PTH [34]. Therefore, harmonisation of the PTH assay remains the most pragmatic solution for wider adoption and the IFCC has created a working group for PTH [35].

A recent study involving four British laboratories, which all use the same assay for measurement of PTH, found that locally defined reference ranges using an indirect method were very different in their respective populations, and were very dissimilar from the manufacturer’s reference range. This meant that use of the manufacturer’s reference range would cause potential diagnostic discordance [36]. Moreover, PTH concentration increases with age independent of 25-hydroxyvitamin D, renal function, ionised calcium, and phosphate [37,38]. Most laboratories currently do not use age-specific PTH reference intervals and may be over-diagnosing parathyroid disorders in the elderly. Recently, age-specific PTH reference intervals derived using indirect methods have become available for commonly used methods [39,40,41].

### 3.6. Glucocorticoid Deficiency and Excess

Immunoassays were the method of choice for routine cortisol quantification in serum in British medical laboratories as recently as 2017 [42], and most laboratories continue to use immunoassay as first line serum cortisol method. The analysis of cortisol is not free from the issues seen with other hormones. A study by Clark et al. [43] published in 1998 showed that four different immunoassay platforms produced varying lower cortisol reference limits for the response in healthy controls to a standard short Synacthen test. At 30 min post-Synacthen dose, the fifth centile for cortisol in 100 healthy controls ranged from 510 to 626 nmol/L [43]. Since then, the major immunoassay manufacturers have adjusted their assays to be more consistent with the LC-MS/MS method. However, drawing evidence from EQA reports, a study in 2017 reported significant differences in serum cortisol results from different immunoassays and suggested using assay-specific cut-offs for interpretation of post-ACTH cortisol in the short Synacthen test [42].

The investigation of possible cortisol excess (Cushing’s syndrome and mild autonomous cortisol excess) is also affected by variation in assay performance. The Endocrine Society’s guideline recommends using one test out of an overnight dexamethasone suppression test, a 48 h low dose dexamethasone suppression test, at least two late night samples for salivary cortisol, or at least two urine collections for 24 h urine free cortisol measurement as the initial screen for Cushing’s syndrome [44]. Interpretation of the urine free cortisol is recommended to be guided by the laboratory’s reference range. However, the overnight dexamethasone suppression test is to be interpreted as positive if serum cortisol is >50 nmol/L and the midnight salivary cortisol is positive if >4 nmol/L. The guideline does go on to admit that there is variation in assay performance, which may lead to incorrect interpretations using these values. To reduce the risk of false positives, the Endocrine Society’s guideline states that positive results on an initial test for excess cortisol must be confirmed with a second different test from the list above before proceeding with further investigation [44].

In a study that highlighted the difficulty of diagnosing Cushing’s syndrome, 156 obese patients gave midnight salivary samples that were analysed using both radioimmunoassay (RIA) and LC-MS/MS [45]. The investigators used their methods’ upper limit of normal for salivary cortisol (both determined in healthy volunteer populations). For RIA it was 4.7 nmol/L and for LC-MS/MS it was 2.8 nmol/L, highlighting that use of a single cut-off as suggested by the Endocrine Society would not be possible for salivary cortisol measurement. Secondarily, the RIA gave a mean result of 4.06 nmol/L and those by LC-MS/MS had a mean value of 1.13 nmol/L. In this analysis of saliva using these two methods, interpretations were in agreement in only 88% of patients when the method-specific decision threshold was used. Using a standard value such as the 4.0 nmol/L recommended by the Endocrine Society [44] would have led to marked discordance. Also of interest was the fact that no patients who had a positive result for midnight salivary cortisol via either method in this study went on to be diagnosed with Cushing’s syndrome. In this study, the investigators repeated the experiments with paired measurements from 50 non-obese volunteers. There was similar overall agreement (92%) between the two methods. Obesity is a known cause of non-Cushing’s hypercortisolaemia. However, the false positive rates were similar in both groups; by RIA it was 15% in obese vs. 18% in non-obese, and by LC-MS/MS it was 8% in obese vs. 14% in non-obese [45].

The 50 nmol/L serum cortisol cut-off for a positive test after an overnight dexamethasone suppression test is not immune to analytical variability. In a study in which samples from dexamethasone suppression tests were analysed for cortisol concentration using three different immunoassays (Roche, Abbott, and Siemens) and then compared with LC-MS/MS as a standard, the assays showed variation in performance [46]. The Abbott method had the most significant negative bias compared to the LC-MS/MS method. The authors found 19.2% of patients would inappropriately not go on to have a second biochemical investigation due to a “normal” (<50 nmol/L) result if the Abbott method was used compared with the result from LC-MS/MS. The authors concluded that users of the Abbott method should consider lowering their cut-off value for a positive result to 31.2 nmol/L [46]. Roche’s method was also negative compared with LC-MS/MS, but much less so, and the Siemens platform gave a slight positive bias [46]. The almost universal use of 50 nmol/L as a discriminator for these overnight dexamethasone suppression tests without consideration of this variation in performance is possibly causing variation in decision making about who needs a second biochemical test when a diagnosis of Cushing’s syndrome is being considered.

### 3.7. Diabetes Mellitus

A1c fraction of the irreversibly glycated haemoglobin A (HbA1c) is used in the diagnosis and monitoring of diabetes mellitus. In common with other analytes, there exist multiple ways of measuring it, including immunoassays, enzymatic methods, ion exchange chromatography, capillary electrophoresis, and high-performance liquid chromatography. Diabetes mellitus, in its various forms, is common. As a result, a lot of work has been undertaken, initially by the National Glycohemoglobin Standardization Program (NGSP) and then by the IFCC, in the standardisation of the measurement of HbA1c internationally. All manufacturers now should have their methods calibrated to the standards created by the IFCC’s reference measurement procedures in 1 of 15 international laboratories [47]. This has improved variation in HbA1c measurements and in some ways this test is an example of international collaboration successfully improving harmonisation. Yet, it has not removed variation in its entirety. The first EurA1c trial included 2166 laboratories across 17 countries in Europe and covered methods made by 24 manufacturers. Each lab had to test the same two levels of HbA1c in either whole blood or lyophilised haemolysates [48]. This showed that, overall, the laboratories had low bias in their reporting. However, lab-to-lab variation was still present. Some countries performed better than others and similarly some methods outperformed the rest. They concluded that the state of the art for HbA1c measurement in Europe was such that “if HbA1c of a patient is measured in 1 of the 2166 laboratories that participated in the EurA1c trial, it can be expected that 1 of 20 laboratories will report a result that will differ 5 mmol/mol (0.46%) or more from the true value” [48]. A 5 mmol/mol difference may be enough that a non-diabetic is labelled as having diabetes mellitus or vice versa. This cannot be ameliorated with locally derived reference ranges, as can be done for other analytes, because the HbA1c cut-off for diagnosis of type 2 diabetes mellitus (≥48 mmol/mol) is set internationally by the World Health Organisation [49]. The EurA1c trial is now in its seventh iteration and the reports are available online for the preceding six trials [50]. In the latest available report (2021 trial), bias by manufacturer in the lyophilised haemolysates ranged from +5.0 to –7.2 mmol/mol. The range of bias in whole blood was smaller. This suggests the work of standardisation of HbA1c is not yet complete.

### 3.8. Human Chorionic Gonadotropin

Though not typically used in the diagnosis of endocrine disorders, complexities in the measurement of the hormone human chorionic gonadotropin (hCG) deserve mention. HCG is used in the diagnosis and monitoring of pregnancy, trophoblastic disease, and various malignancies. It circulates in many forms in the blood. These include the total molecule comprising an α (common to all glycoproteins) and a β chain, covalently linked and having variable glycosylation; free β chains; “nicked” (partially degraded) β chains or “nicked” whole hCG; and “core” β chains (two particular residues of the β-chain bound together by a disulphide bridge). Additionally, different sites of production in the body can lead to differing amounts of glycosylation of the glycopeptide. It can be of placental origin, from trophoblastic disease, or other malignancies, and can originate from the pituitary gland [51,52,53]. Laboratories should be clear about which chain, fragment, or molecule they are measuring and reporting, and the limitations of measuring that particular molecule [53]. Indeed, this is an important source of variation as different manufacturers have created different hCG assays to serve different purposes, e.g., assays specific for the β chain of hCG are very useful for monitoring pregnancy and as part of the antenatal screening for Trisomy 21 in the foetus, but due to this specificity for the β chain, probably do not function as well in the role of tumour markers as those assays that can detect all the fragments and the whole molecule [53]. Clinicians requesting and interpreting these tests should know this information, but it may not always be clear to them. Assay discordance may also arise because the immunoassays react variably to different fragments or isoforms in unintended ways [53,54,55,56] such that two manufacturers who both aimed to measure only one isoform have different levels of cross-reactivity by other fragments or isoforms. This lack of harmonisation among different hCG assays, even of the same isoform or fragment, can be a source of diagnostic and treatment error. A study assessing congruity between seven major hCG assays showed biases ranged from +30.9 to −37.5% when the WHO fourth international standard for hCG was used, and from +36.8 to −36.1% when patient samples were used [57].

HCG can be slightly elevated in women in the peri- and post-menopausal periods [58,59,60,61], and may also be detected in healthy men [62]. Most literature cites the pituitary as the source of the small amounts of hCG in the menopausal period [58,59,60,61]. However, others have historically hypothesised that lack of assay specificity leads to this slightly raised hCG level through immunoassay binding of glycosylated luteinizing hormone (LH) molecules (which are also raised in the post-menopausal period) instead of to hCG [63], although this is less of a concern with modern assays [53]. HCG levels above the commonly used reference range of ≤5 IU/L for non-pregnant women can be found in up to 10.6% of post-menopausal women who are not pregnant and have no disease or tumour, with some post-menopausal women having levels as high as 10–14 IU/L without pathology [58,61]. This can have significant effects on clinical decision making, as raised hCG levels may prompt concerns of trophoblastic disease, tumours, or pregnancy, and may therefore expose patients to unnecessary investigations and potentially harmful therapies including chemotherapy, as occurred in 6 out of 36 peri/post-menopausal women with raised hCG in one case series [60]. It is a strong argument for the need for age- or menopausal-status-specific reference intervals for hCG in women.

While assay inaccuracy encountered in its use for the monitoring of pregnancy is mitigated by serial hCG measurements during the pregnancy using the same assay, single hCG measurements that are slightly elevated in a peri/post-menopausal woman do not provide this option. When monitoring patient response to treatment in the case of malignancy, again the same assay should be used each time.

We summarise the above examples of the impact of measurement variation in Table 1 below.

## 4. How to Assess for Potential Clinical Decision Discordance Resulting from Between-Assay Differences?

Regression analysis and difference plots are routinely used by laboratories worldwide for numerical comparison of results from different methods, in particular when a new assay is introduced, or the method is changed [64]. However, differences in clinical interpretation because of different method-specific reference intervals, when similar results are generated by two methods, are not assessed routinely. When there are differences between results of different methods, it is generally believed that they are compensated for by method-specific reference intervals, typically provided by the manufacturer [65]. However, as evident from the studies summarised earlier, this may not always be the case and significant clinical management discordance may get overlooked in routine method comparison studies. Moreover, assays can change over time, invalidating the initial method comparison studies. Assay drift, which is the technical term for unintentional, usually gradual, changes in the assay measurement performance over time, is a factor that should also be considered. It can be a potential cause for incongruent results across assay platforms but also for samples tested in the same lab using the same method at different times. Laboratories usually have processes in place to detect and reverse it when it does occur, including regular internal quality control (IQC) tests and subscription to external quality assurance (EQA) schemes. Well-recognised rules exist for interpreting the results of IQC tests, such as the Westgard rules [66]. These and other similar rulesets are meant to help the laboratory scientist detect significant changes in the assay performance, while minimising the investigation of random but acceptable assay fluctuations. These rules applied to the analysis of IQC have done wonders for allowing a busy clinical laboratory to keep an eye on performance while still providing a high-throughput service. However, there are questions about whether the traditional Westgard rules are the best for detecting drift [67,68]. There will always be a trade-off between detecting assay drift as early as possible, on one hand, and reducing time wastage caused by investigating more changes in IQC than necessary on the other.

EQA schemes are also of proven value in identifying gradual changes with time. However, it is important to be cautious because EQA samples may not be fully representative of clinical samples due to analyte or matrix-related differences. A recent study demonstrated that samples spiked with pituitary-derived TSH, often used by EQA schemes, did not fully reveal the extent of differences between Abbott’s and Roche’s TSH assays [69]. This was believed to be because of glycosylation or other isoform differences between serum TSH and pituitary-derived TSH [69]. Manufacturers also have a role to play in monitoring and responding to changes in the performance of the reagent kits that they provide to medical laboratories, and should engage with and support harmonisation and standardisation initiatives.

It is pragmatic to consider clinical discordance checks on an analyte-by-analyte basis, especially when signalled by clinician feedback. The traditional method employed in assessment of clinical discordance involves identification of condition-specific samples, simultaneous analysis of the samples using the methods of interest, interpretation of results using the respective method’s specific reference intervals or against clinical decision thresholds, and preparation of discordance tables or discordance plots to compute the proportion of individuals for whom the clinical management would have been affected [16,24,70]. This traditional clinical concordance assessment experiment, however, may not be feasible for large numbers of analytes in busy clinical laboratories. A recent study used Passing–Bablok regression equations derived using routine method comparison studies for regression transference of readily available routine clinical data and thereafter interpretation using method-specific reference intervals to identify potential clinical discordance between methods [23]. Using the example of clinical management discordance in subclinical hypothyroidism via assays for TSH and free T4 from Abbott and Roche, this study demonstrated that the regression-based approach using real-world clinical data identified a similar magnitude of clinical management concordance/discordance as was found in resource-intensive sample exchange studies. This regression-based method could be partially automated, making it less resource intensive and therefore better suited to busy clinical laboratories.

## 5. Mitigating the Impact of Assay Differences on the Management of Patients

Reference intervals or decision thresholds specific for the population or clinical question can be derived by direct or indirect methods, or by transference. They will make interpretation of results more accurate in the clinical laboratory, but deriving them may not always be feasible or possible [71]. Even if the assays are harmonised or standardised, it is unlikely that reference intervals addressing every possible factor affecting test results will ever be available; examples of these factors are intra-individual circadian and seasonal variation, inter-individual differences, age-, sex-, and ethnicity-related differences, and the effect of diet, physical activity, other medical conditions, and medications. The most specific available reference interval or decision threshold for the situation should be used and, more importantly, it is imperative to “treat the patients and not numbers” and individualise decisions based on the complete clinical picture [72]. Medical laboratories can also refer samples to reference laboratories if there is any doubt about the accuracy or interpretation of the results produced. Close links with, and free-flowing feedback from, requesting clinicians will help to highlight the cases where this may be needed.

Although alleviation of assay-related differences in the diagnosis is challenging, it is relatively straightforward in the follow-up monitoring of patients. Using the same assay for follow-up measurements largely mitigates assay-related factors. However, occasionally undetected gradual cumulative assay drift over months to years may cause clinically significant variation, as discussed before.

## 6. Conclusions

Variation in the performance of assays used in endocrinology investigations exists across many analytes and can play a significant role in diagnostic uncertainty or errors. Variability arises from multiple origins and can be difficult to detect in day-to-day clinical work. Continued national and international efforts at harmonisation and standardisation will hopefully improve this situation in the future. However, individual medical laboratories that provide measurement of hormones for the diagnosis and monitoring of patients have their role to play in implementing robust IQC processes and taking part in EQA. Manufacturers of the assays should implement robust quality assurance practices to minimise lot-related changes and assay drifts. The manufacturers should engage with harmonisation and standardisation efforts and implement resulting changes to the assays. Locally derived reference intervals are important, and where possible, should be as specific to the clinical and demographic characteristics of the patient as possible, although we acknowledge the difficulties with obtaining these. Long term follow-up of patients using the same assay makes decision making more straightforward. The laboratory should have close links with the clinicians who use its services so that they can keep them abreast of analytical challenges and can receive feedback from them. Variation has improved in biochemical assessment of endocrinological disorders but much is still to be done.

## Figures and Tables

**Table 1 diagnostics-13-03453-t001:** Impact of assay and reference interval differences on the diagnosis and management of endocrine conditions.

Endocrine Condition (s)	Examples of Laboratory Variation Impacting Diagnosis/Monitoring	Reference (s)
Growth Hormone Deficiency and Excess	IGF-1 measurement on different platforms can give different results.	[5,6]
	IGF-1 reference intervals derived from the same local population using different platforms can be different.	[5]
	Manufacturer-provided reference ranges for IGF-1 can lead to differences in interpretation of results when compared to reference intervals derived from local population.	[7,8,9]
	Measurements of IGF-1 and GH (after GH dynamic function tests) can be incongruent due to assay performance issues.	[5,6,12,13,14]
Hypothyroidism and Subclinical Hypothyroidism	Biases in TSH and fT4 measurement on major immunoassay platforms, with reference ranges that enhance rather than mitigate the discrepancy, can lead to differences in clinical decision making.	[20,21,22,23]
	Only 44% of 53 samples had same diagnosis of subclinical hypothyroidism on two different immunoassay platforms in one study.	[24]
	There was 14% potential difference in decision making when monitoring 100 patients taking levothyroxine for hypothyroidism by two different immunoassays.	[25]
PCOS	Measurement of testosterone in women using immunoassays is associated with poor accuracy and precision and thus a risk of misdiagnosis when evaluating for PCOS.	[26]
	Some reference intervals for testosterone in women are likely to have been derived from women with PCOS and so may be too wide and lead to mislabelling of women with PCOS as being disease free.	[26]
Hypogonadism	Lower reference limits for testosterone in men vary widely across laboratories, with relatively few of these being derived from local populations. Thus, detection rates of male hypogonadism could vary from laboratory to laboratory.	[27]
	Large biases from target concentrations have been found on some immunoassays for testosterone levels in EQA schemes and this could cause variation in detection rates of male hypogonadism.	[27]
	Most modern methods do not measure low concentrations of oestradiol with good precision.	[29]
	Assay-related variation in oestradiol results can change diagnosis rates of ovarian failure across laboratories.	[30,31]
Parathyroid Disorders	PTH results for a specimen can vary significantly across platforms. Only 29% of 55 samples with a normocalcaemic hyperparathyroidism on one platform had the same diagnosis on two other platforms in a study.	[33]
	PTH reference intervals may vary dependant on population. Therefore, using the manufacturer-provided reference interval, which many laboratories do, may result in wrong classification of PTH results.	[36]
	PTH rises physiologically with age, but many laboratories use the same reference interval for all adults and therefore may erroneously classify a PTH result as elevated in the elderly.	[37]
Glucocorticoid Deficiency and Excess	Laboratories may have to derive their own cut-off for post-Synacthen cortisol result because of differences in cortisol assays.	[42,43]
	Endocrine Society guidance considers a late-night salivary cortisol result >4 nmol/L indicative of hypercortisolism. However, a locally derived reference range for midnight salivary cortisol in healthy adults in one study was <4.7 nmol/L by RIA and <2.8 nmol/L by LC-MS/MS.	[44,45]
	The overnight dexamethasone suppression test is used as one of the initial investigations for glucocorticoid excess with guidance that a positive result occurs when cortisol is >50 nmol/L. However, variation in cortisol assay performance causes differences in interpretation using this cut-off if different assay platforms are used.	[44,46]
Diabetes Mellitus	Among the 2166 European laboratories in the first EurA1c trial, 1 in every 20 laboratories could report a value for HbA1c that differed from the target by ≥5 mmol/mol and in the seventh iteration of the trial bias could be as high as −7.2 mmol/mol. These biases may be enough to cause misclassification of a patient’s diabetes status.	[48,50]
Human Chorionic Gonadotrophin	This hormone, though not usually implicated in pathology of the endocrine system, can show profound assay-related differences. A study of seven major hCG assays showed biases that ranged from +30.9 to −37.5% when the WHO 4th international hCG standard was assayed.	[57]
	Upper reference intervals for hCG for patients who are not pregnant may be too restrictive when applied to post-menopausal women and have led to inappropriate investigation and treatment in this group.	[60]

## Data Availability

Not applicable.

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
