# Peer review of "Impact of Variation between Assays and Reference Intervals in the Diagnosis of Endocrine Disorders"

_diagnostics, 2023, doi:10.3390/diagnostics13223453_

Round 1
Reviewer 1 Report
Comments and Suggestions for Authors
This study is devoted to a relevant topic. Factors influencing the results of laboratory tests are considered. The review is well written, has a clear structure, and uses up-to-date sources.
General comment: the authors provide in the text of the article several strategies for assessment and mitigation of the discordance, which are not mentioned in the conclusion. Among them: the need for age specific reference intervals, the need to inform the clinician about the features and limitations of the analysis, and the need for clinician feedback to laboratories.
Minor comments:
Line 139: “LC-MS/MS methods” ˗ Abbreviation expansion
Line 152: “(+ Kalaria T et al- in press)” ˗ Add link or remove
Line 164: “test. [32].” ˗ “test [32].
Line 214: “HPLC” ˗ Abbreviation expansion
Line 271: “LH molecules” ˗ Abbreviation expansion
Line 495: “lancet Diabetes” ˗ Lancet Diabetes
Author Response
Dear Reviewer,
Thank you for taking the time to review our manuscript and for your comments and recommendations:
"General comment: General comment: the authors provide in the text of the article several strategies for assessment and mitigation of the discordance, which are not mentioned in the conclusion. Among them: the need for age specific reference intervals, the need to inform the clinician about the features and limitations of the analysis, and the need for clinician feedback to laboratories."
We have made amendments to conclusion (lines 378-383)
"Line 139: “LC-MS/MS methods” ˗ Abbreviation expansion"
Abbreviation expansion of LC-MS/MS- abbreviation expanded on lines 112-113.
"Line 152: “(+ Kalaria T et al- in press)” ˗ Add link or remove"
This text has been removed
"Line 164 (now 165 -166): “test. [32].” ˗ “test [32]."
Amendment made as advised
"Line 214 (now 218): “HPLC” ˗ Abbreviation expansion"
Amendment made as advised
"Line 271 (now 277): “LH molecules” ˗ Abbreviation expansion"
Amendment made as advised
"Line 495 (now 549): “lancet Diabetes” ˗ Lancet Diabetes"
Amendment made as advised.
Yours sincerely,
Corresponding Author
---------------------------------------------------------------
Reviewer 2 Report
Comments and Suggestions for Authors
Dear authors,
Your review represents a huge work and rises an important issue which is the impact of variation in assays used to diagnose endocrine disorders. I agree with the observation that, depending to the test (company) used, the results can be different. Thus, it is important to summarize knowledge to rise the problem. And I thank you for that.
However, I disagree with one idea and I think that your literature review forgot one point.
I disagree with the general idea, and even if it is not clearly mentioned in your review, it seems to be your opinion, that one test is not only use to propose a diagnosis. For instance, when you write "The almost universal use of 50 nmol/L as a discriminator for these overnight dexamethasone suppression tests without consideration of this variation of performance is likely causing variation in decision making about who to further evaluate when a diagnosis of Cushing’s syndrome is being considered.", it seems that you have the idea that the decision of further investigation will be taken only with the data of the overnight dexamethasone suppression test. It is generally not the case. First, there is a clinical suspicion. Secondly, at least two tests will be realized (Late-night salivary cortisol, and/or dexamethasone suppression test, and/or 24-h urine-free cortisol, and/or midnight serum cortisol measurements). The most often, at least two tests are realized because physicians are aware of difficulties in results interpretation.
To my point of view, you forgot, the evolution of the clinical diagnosis with the reference centers. Now, and more and more, pathologies are studied in specialized centers. In routine, in case of doubt of a diagnosis serum or plasma, or urine from patient are send to this reference centers. Here, we avoid the difference observed in the methods used for quantification. One sentence has to be written in this sense.
I have 3 last comments:
1/ Line 154: "Immunoassays remain the commonest method for routine cortisol quantification in serum". Please reference your sentence. To the best of my knowledge, the commonest method is LC-MS.
2/ Line 152: I am not sure at all that it is correct to cite results which are not yet published in a review.
3/ In conclusion: "However, laboratories have their individual roles to play by implementing robust IQC processes and taking part in EQA". I agree if you are talking about laboratories in hospital but also the ones from kit providers. Please, be more precise in your idea.
Author Response
Dear Reviewer,
Thank you for taking the time to review our manuscript and for your comments and advice:
"However, I disagree with one idea and I think that your literature review forgot one point.
I disagree with the general idea, and even if it is not clearly mentioned in your review, it seems to be your opinion, that one test is not only use to propose a diagnosis. For instance, when you write "The almost universal use of 50 nmol/L as a discriminator for these overnight dexamethasone suppression tests without consideration of this variation of performance is likely causing variation in decision making about who to further evaluate when a diagnosis of Cushing’s syndrome is being considered.", it seems that you have the idea that the decision of further investigation will be taken only with the data of the overnight dexamethasone suppression test. It is generally not the case. First, there is a clinical suspicion. Secondly, at least two tests will be realized (Late-night salivary cortisol, and/or dexamethasone suppression test, and/or 24-h urine-free cortisol, and/or midnight serum cortisol measurements). The most often, at least two tests are realized because physicians are aware of difficulties in results interpretation. "
- We have edited lines 177 to 180 as well as lines 205 to 207 and 212 to 213 to make it clearer that the Endocrine Society guideline does advise that a second biochemistry test must be done if the first screening test for excess cortisol is positive and this second test acts as a mitigator against the risks of false positive results in the first test chosen.
"To my point of view, you forgot, the evolution of the clinical diagnosis with the reference centers. Now, and more and more, pathologies are studied in specialized centers. In routine, in case of doubt of a diagnosis serum or plasma, or urine from patient are send to this reference centers. Here, we avoid the difference observed in the methods used for quantification. One sentence has to be written in this sense."
- We have added lines 361 to 364 to highlight the benefit of reference laboratories.
"1/ Line 154: "Immunoassays remain the commonest method for routine cortisol quantification in serum". Please reference your sentence. To the best of my knowledge, the commonest method is LC-MS."
-
We have edited and added a reference to show that we specifically meant that immunoassays have been chosen more often than not by British medical laboratories for quantification of cortisol in serum. References 31 and 32 have been rearranged ( reference to this statement is now reference number 31 but was previously number 32)
"2/ Line 152: I am not sure at all that it is correct to cite results which are not yet published in a review. "
-
Line 152- Reference to unpublished work removed
"3/ In conclusion: "However, laboratories have their individual roles to play by implementing robust IQC processes and taking part in EQA". I agree if you are talking about laboratories in hospital but also the ones from kit providers. Please, be more precise in your idea. "
-
We have edited lines 378 to 383 to be more precise in meaning as advised. We have also added lines 327-330 to mention the responsibility of kit manufacturers.
Thank you again.
Yours sincerely,
Corresponding author